# Changes in a Fish Community in a Small River Related to the Appearance of the Invasive Topmouth Gudgeon *Pseudorasbora parva* (Temminck & Schlegel, 1846)

**Jacek Rechulicz** 

Department of Hydrobiology and Protection of Ecosystems, University of Life Sciences in Lublin, Dobrzańskiego str. 37, 20-262 Lublin, Poland; jacek.rechulicz@up.lublin.pl; Tel.: +48-81-4610061 (ext. 321)

**Abstract:** In recent years, the topmouth gudgeon (*Pseudorasbora parva*) has been one of the most invasive fishes in Europe. *Pseudorasbora parva* can potentially affect ecosystems, fish communities, and particular fish species. Electrofishing was carried out over a five-year period at three study sites in the Ciemięga River (eastern Poland) before and after *P. parva* had been found in the river. Changes in the occurrence, abundance, and density of native fish species after the appearance of the invader were determined. Changes in the species' richness were calculated, and correlations were estimated between the occurrence and density of *P. parva* and particular fish species and richness indices. The presence of *P. parva* has not affected the density of native species but coincides with a significant increase in estimated species richness and the total density of fish. There was a significant relationship between the presence of this invasive species and the fish community's composition, though the PERMANOVA result was unclear with regards to site-specific effects. Moreover, tench and bleak were associated positively, whereas Eurasian perch and sunbleak were associated negatively with the occurrence of *P. parva*. *Pseudorasbora parva* density was highly correlated with Cyprinidae density (excluding *P. parva*), species richness, and the Margalef diversity index. Thise study has shown that the presence of predatory fish in the river (*Salmo trutta* L.) may reduce the numbers of invasive *P. parva*.

**Keywords:** *Pseudorasbora parva*; invasive fish species; native fish; river; fish diversity

## 1. Introduction

Over the last twenty years, the topmouth gudgeon *Pseudorasbora parva* (Temminck et Schlegel, 1846) has been one of the most invasive fish species in Europe [1], including the UK, Spain, and Turkey [2–5]. These studies have often focused on identifying the characteristics of *P. parva* that enable its successful invasion. Some studies on *P. parva* have highlighted such successful features as its high morphological plasticity [6], its early age of sexual maturity and fertility [7], and its opportunistic feeding strategy [8–13]. Moreover, the introduction pathways and problems caused by *P. parva*, such as habitat utilisation, infection, and disease, have been reviewed [14–18].

Invasive species are known to have a potentially significant negative impact on ecosystems and freshwater diversity [19–21]. Further, *P. parva* may affect the volume of carp production in ponds [22], compete for food with native species [8,10,23,24], and influence their reproduction [25]. Certain ecosystem properties exert protective effects on species invasions. One such mechanism is the presence of predators, which can effectively eliminate single individuals of an alien species [26]. Inherent issues in determining the effect of a non-native species is that few ecosystems are monitored pre-invasion [27,28].

The present study provides a unique opportunity to determine the relationships between *P. parva* and the fish community and fish diversity in a small river, with observations carried out before and after its invasion. The aims of this study were to determine (i) whether the appearance of *P. parva* corresponded with changes in the abundance and density of native fish species; (ii) whether the appearance of *P. parva* coincided with the changes in selected species richness indices; and (iii) whether the occurrence and numbers of this invasive species were related to the presence of predatory fish.

## 2. Materials and Methods

This study was conducted in the River Ciemięga (36 km long, 157 km² catchment basin) which is a small river in south-eastern Poland, where *P. parva* was first found in autumn 2005 [29] (Figure 1). Three 100 m long sections of the river were sampled (study sites; S1, S2, and S3) (Table 1). Between site S2 and S3, upstream from the village of Pliszczyn, there is a several metre-long weir preventing the upstream migration of fish. Sites S1 and S2 are typical for an intermediate lowland-upland river. The river section beyond the weir (S3; near the village of Pliszczyn) is a trout and grayling zone typical for a mountain river. During the research, the physicochemical parameters of water at all study sites were measured in situ using a multiparameter meter YSI 556 (YSI Inc, Yellow Springs, OH, USA). The characteristics of the study sites and water parameters are shown in Table 1.

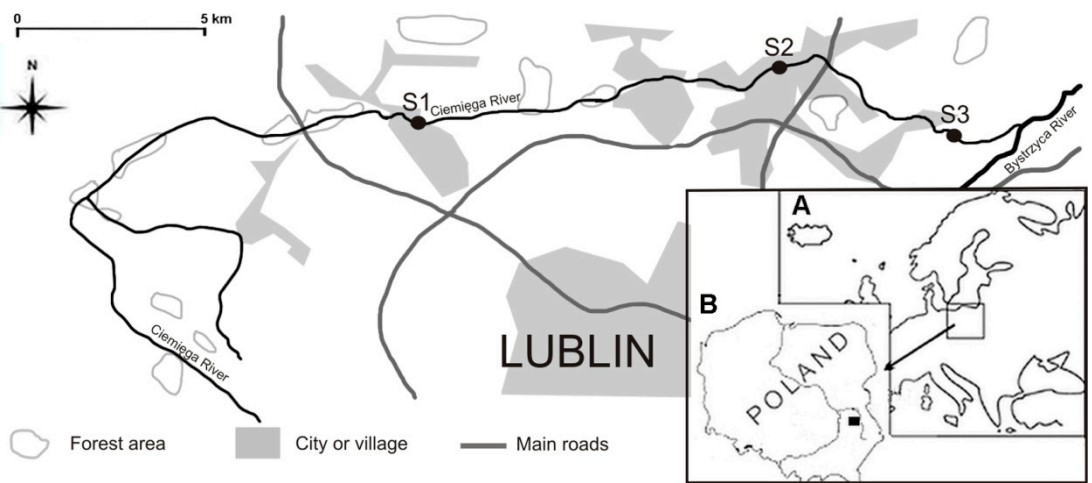

**Figure 1.** Map showing the location of study area in Europe (A) and Poland (B) and study sites on the Ciemięga River (S1–S3).

**Table 1.** Characteristics and physicochemical water parameters (mean ± SD)† at sampling sites in the Ciemięga River (after Rechulicz [29], altered).

| Site Specification | S1 Jastków | S2 Dys | S3 Pliszczyn |
|---|---|---|---|
| GPS site coordinates | N 51°18′36″ E 22°26′00″ | N 51°18′49″ E 22°33′52″ | N 51°18′08″ E 22°38′28″ |
| Distance from the river spring (km) | 18 | 26 | 33 |
| Width of the river bed (m) | 2.50–3.00 | 3.50–4.00 | 4.50–6.00 |
| River depth (m) | 0.30–0.60 | 0.40–0.90 | 0.30–0.45 |
| Bottom characteristics | Sand and gravel bottom, many submerged macrophytes | Sand and gravel bottom, in some sections mud, many submerged macrophytes | Gravel and sand bottom, few macrophytes |

**Table 1.** *Cont.*

| Site Specification | S1 Jastków | S2 Dys | S3 Pliszczyn |
|---|---|---|---|
| River bed characteristics | Straight, flat banks, low banks with plants: *Urtica dioica L, Lamium album* L., a few trees | Irregular with a small meander, high and steep banks with plants: *Urtica dioica L, Lamium album* L., a few trees | Shallow, straight, flat banks, high valley, trees along the banks: *Alnus glutinosa, Salix alba* L. |
| River modifications | Partially regulated | Partially regulated, banks with concrete blocks near a bridge | Natural river, without modifications, with a weir above the study site |
| pH | 7.5 ± 0.3 | 7.57 ± 0.32 | 7.5 ± 0.5 |
| Conductivity ($\mu S\ cm^{-1}$) | 611.2 ± 159.2 | 638.6 ± 49.6 | 581.4 ± 150.8 |
| Temperature (°C) | 11.2 ± 5.8 | 10.8 ± 5.0 | 10.2 ± 3.7 |
| Oxygen saturation (%) | 80.0 ± 6.9 | 79.5 ± 25.0 | 93.1 ± 5.9 |
| Dissolved oxygen ($mg\ L^{-1}$) | 8.2 ± 1.4 | 8.9 ± 1.9 | 9.3 ± 2.0 |

[†] values for the period from spring 2006 to autumn 2007.

Observations were conducted at each study site three times before the recording of *P. parva* (on dates: 6 October 2003, 4 October 2004, and 9 May 2005) and seven times after its appearance (on dates: 3 October 2005, 8 May 2006, 1 August 2006, 2 October 2006, 7 May 2007, 2 August 2007, and 1 October 2007). Electrofishing was performed using electric gear (IUP-12, 220-250V, 7A, Radet, Poznań, Poland) across the entire width of the riverbed while wading upstream [30]. Before the inventory fishing, the upper and lower parts of each site were separated by a net barrier with a mesh size of 8 mm. The collected fish had their species identified and were weighed (W; to the nearest 0.1 g), and their total lengths (Tl) were measured (to the nearest 1 mm). Native fish specimens were put back in the water, and all non-native fish specimens were killed (in accordance with the applicable national ethical regulations) with an overdose of 2-phenoxyethanol and then immediately preserved in 4% formaldehyde.

For each fish species on all the sampling dates, and for the period before and after the appearance of *P. parva*, the dominance index (n%) was calculated as:

$$n\% = 100 \times n_i/N \tag{1}$$

where $n_i$ is the number of individuals of species i and N is the total number of fish. Moreover, the density of all fish species was estimated as a catch per unit effort (CPUE), which was the number of fish per 100 $m^2$ (ind. $\times 100\ m^{-2}$).

Native fish species were classified into three groups: predatory fish, Cyprinidae (in all subsequent analyses and reports, this group was considered excluding *P. parva* itself), and other fish. The predatory fish category, i.e., the brown trout (*Salmo trutta* L.), northern pike (*Esox lucius* L.), and Eurasian perch (*Perca fluviatilis* L.), consisted of species that may limit the number of *P. parva*. The second group, fish of the cyprinid family (which includes *P. parva*) may be affected by competition for habitat and food. The third group consisted of the stone loach (*Barbatula barbatula* L.) and threespine stickleback (*Gasterosteus aculeatus* L.) as non-predatory and non-cyprinid fish species.

For each sample, the number of species (species richness, S) was calculated and the jackknife 1 procedure [31] was used to assess the estimated species richness (eS) via the following formula:

$$eS = S_{obs} + Q_1(m - 1)/m \tag{2}$$

where $S_{obs}$ is the number of species in the sample, $Q_1$ is the number of species that occur in one sample only (unique species), and m is the number of samples.

Moreover, the rarefaction technique, assessing a theoretical number of taxa that would be found in a given number of collected individuals [32], was used to control for different abundances of fish collected at various sites and on particular dates. Additionally, the Margalef diversity index was calculated using the following formula:

$$R = S - 1/\ln(N) \tag{3}$$

where S is the number of species and N is the total number of observed individuals.

In addition, the turnover rate of fish (*t*) was calculated according to the formula by Brown and Kodric-Brown [31]:

$$t = (b + c)/(S_1 + S_2) \tag{4}$$

where b is the number of fish species present only in the period before *P. parva* was noted, c is the number of fish species present only in the period after *P. parva* was noted, $S_1$ is the total number of species found before *P. parva* was noted, and $S_2$ is the total number of species present after *P. parva* was noted. This index ranges from 0 to 1 (where 0 indicates no difference in the fish fauna and 1 means the complete replacement of the fish fauna).

Prior to the statistical analysis of fish density (each species separately and all species pooled) and richness (estimated species richness, species richness rarefacted for 20 individuals and Margalef diversity index), the data were checked for normality (Shapiro-Wilk test) and homogeneity of variances (Levene's test). Due to strong violations of these assumptions, a non-parametric Mann–Whitney U test was used to compare these variables between the periods before and after the appearance of *P. parva*. Non-parametric Spearman rank correlations were calculated to check the relationships between the density of *P. parva* and the percentage abundance and density of each group of fish, as well as the richness indices (using only the data obtained after the appearance of *P. parva*). These analyses were carried out with the StatSoft Statistica v. 10 software [33], with a significance level of $p \leq 0.05$.

To reveal the relationships between species, sites, and environmental variables, a canonical correspondence analysis (CCA) was conducted. Log-transformed fish densities were used as species data, except for the density of *P. parva* (also log-transformed), which was included as one of the environmental variables potentially affecting the fish assemblage. Rare species (represented by less than five individuals) were removed from the dataset to reduce noise. The significance of the relationships between fish species and environmental variables along particular CCA axes, as well as the significance of each environmental variable contribution, were tested using permutation tests with 1000 replicates.

The difference in the taxonomic composition of fish assemblages before and after the appearance of *P. parva* was tested using a 2-way PERMANOVA (with 1000 replicates) based on a Bray–Curtis distances, with site (1–3) and *P. parva* presence (before/after) as factors and fish densities (excluding *P. parva* itself) as a dependent variable. A significant interaction between these factors was further tested with three separate 1-way PERNANOVA run separately for each site. The CCA and PERMANOVA analyses were carried out using Vegan 2.4.0 package [34] for R [35].

## 3. Results

In total, 2633 fish were caught, belonging to 15 species and 7 families. The most numerous family was Cyprinidae (9 species). This family included two non-native species, i.e., the invasive *P. parva* and gibel carp (*Carassius gibelio* Bloch 1782). In addition, a single individual of another invasive species, the brown bullhead (*Ameiurus nebulosus* Lesueur 1819), was recorded before the appearance of *P. parva* in the river (Table 2). The controlled fishing after the sampling 4 showed that there were no brown bullhead (*Ameiurus nebulosus*), spirilin (*Alburnoides bipunctatus Bloch 1782)*, and sunbleak (*Leucaspius delineates* Heckel 1843), whereas, apart from *P. parva*, another new species, the ide (*Leuciscus idus* L.), was recorded (Table 2).

An analysis of the changes in total fish density (excluding *P. parva*) showed that despite the increasing trend observed after the appearance of *P. parva*, the difference between both periods was non-significant ($p = 0.11$, Table 3). On the other hand, the total density of fish caught (including *P. parva*) significantly increased from 18.6 to 55.5 CPUE. There was a nearly threefold increase in Cyprinidae density, mainly due to the numerous quantity of bleak (*Alburnus alburnus* L.) (mean 17.0 CPUE), while the gudgeon (*Gobio gobio* L.) population state was reduced by about 41%. Predatory fish density increased by more than threefold after the *P. parva* appearance between dates 2 October 2006 and 1 October 2007, where brown trout, especially at S3, made up a large proportion of the predators (Figure 2). However, the relationships between the presence of *P. parva* and the densities of the above-mentioned groups of fish or individual fish species were statistically insignificant (Table 3).

At each study site, the proportion of cyprinids (excluding *P. parva*) decreased as the share of *P. parva* in total fish abundance increased. This was especially apparent in samples from 2 October 2006 at site S1 (when *P. parva* accounted 58% of fish abundance). On the other hand, an increased proportion of Cyprinidae decreased *P. parva* at this site in all samples in 2007, as well as at S2 on 3 October 2005, 8 May 2006 and 7 May 2007 (Figure 2). At site S3, a large proportion of predatory fish (over 78%) resulted in a markedly lower share of *P. parva* and other cyprinids (Figure 2).

The taxanomic composition of fish assemblages had significant differences among sites, as shown with PERMANOVA analysis, with a significant site and *P. parva* interaction (Table 4). However, the post- and pre-invasion periods assemblages were only significant at Site 2.

The most important environmental variable associated with the taxonomic composition of fish along the first CCA axis was the oxygen concentration ($p = 0.01$) (Figure 3). The high values of this parameter were linked to the presence of the brown trout. The occurrence of the trout was negatively related to those of the other fish species, including *P. parva*. The density of *P. parva* differentiated fish assemblages along the second CCA axis ($p = 0.02$). The tench (*Tinca tinca* L.) and bleak were associated positively, whereas Eurasian perch and sunbleak negatively were associated negatively with the occurrence of *P. parva* (Figure 3b). The samples from site 1 were clearly divided into those collected before and after the appearance of *P. parva*.

The appearance of *P. parva* in the Ciemięga River coincided with a significant increase in the estimated species richness, from a mean of 7.5 to 10.1 (Table 5). However, the numbers of species rarefacted for the equal number of individuals (20) did not differ significantly between both periods (3.3 and 3.7, respectively), indicating that the former difference resulted mainly from unequal numbers of sampled fish. Similarly, the Margalef diversity index did not vary significantly between the periods (1.36 and 1.6, respectively, Table 5).

**Table 2.** Total length (Tl, cm) and biomass (W, g) as the mean ± standard deviation (SD) of fish species in the Ciemiega River; N—number of fish, NS—native, and NNS—non-native species.

| Fish Species | Family | Status | Before Finding *P. parva* | | | After Finding *P. parva* | | |
|---|---|---|---|---|---|---|---|---|
| | | | N | Tl | W | N | Tl | W |
| Gudgeon (*Gobio gobio*) | Cyprinidae | NS | 106 | 11.0 ± 1.4 | 12.4 ± 5.4 | 146 | 11.2 ± 2.8 | 14.2 ± 7.5 |
| Roach (*Rutilus rutilus*) | Cyprinidae | NS | 34 | 12.4 ± 4.2 | 28.5 ± 28.2 | 98 | 11.3 ± 1.8 | 14.2 ± 9.9 |
| Tench (*Tinca tinca*) | Cyprinidae | NS | 1 | 5.9 ± 0.0 | 2.5 ± 0.0 | 34 | 8.1 ± 3.0 | 9.6 ± 13.8 |
| Bleak (*Alburnus alburnus*) | Cyprinidae | NS | 3 | 4.0 ± 0.5 | 0.9 ± 0.2 | 714 | 4.7 ± 0.9 | 1.2 ± 0.4 |
| Spirilin (*Alburnoides bipunctatus*) | Cyprinidae | NS | 1 | 6.8 ± 0.0 | 6.0 ± 0.0 | 0 | - | - |
| Sunbleak (*Leucaspius delineatus*) | Cyprinidae | NS | 7 | 5.8 ± 1.0 | 2.3 ± 1.4 | 0 | - | - |
| Ide (*Leuciscus idus*) | Cyprinidae | NS | 0 | - | - | 4 | 13.1 ± 3.3 | 34.8 ± 19.0 |
| Gibel carp (*Carassius gibelio*) | Cyprinidae | NNS | 31 | 9.0 ± 5.4 | 30.3 ± 42.8 | 79 | 9.0 ± 3.6 | 19.2 ± 22.2 |
| Topmouth gudgeon (*Pseudorasbora parva*) | Cyprinidae | NNS | 0 | - | - | 316 | 6.2 ± 1.0 | 2.2 ± 1.3 |
| Brown trout (*Salmo trutta*) | Salmonidae | NS | 53 | 27.2 ± 3.9 | 235.0 ± 84.9 | 549 | 17.0 ± 5.5 | 67.2 ± 72.8 |
| Northen pike (*Esox lucius*) | Esocidae | NS | 2 | 22.0 ± 1.4 | 62.1 ± 5.5 | 1 | 27.8 ± 0.0 | 151.0 ± 0.0 |
| Eurasian perch (*Perca fluviatilis*) | Percidae | NS | 19 | 10.6 ± 5.3 | 27.0 ± 33.5 | 14 | 9.3 ± 2.1 | 10.5 ± 8.8 |
| Threespine stickleback (*Gasterosteus aculeatus*) | Gasterosteidae | NS | 43 | 5.7 ± 1.2 | 2.3 ± 1.8 | 255 | 4.6 ± 0.9 | 1.4 ± 0.7 |
| Stone loach (*Barbatula barbatula*) | Nemacheilidae | NS | 33 | 8.9 ± 1.9 | 6.5 ± 4.3 | 89 | 9.7 ± 2.2 | 9.1 ± 5.2 |
| Brown bullhead (*Ameiurus nebulosus*) | Ictaluridae | NNS | 1 | 15.0 ± 0.0 | 47.0 ± 0.0 | 0 | - | - |

**Table 3.** Mean (±SD) density (per 100 m²) (CPUE) of native and non-native fish species observed before (autumn 2003–summer 2005; *n* = 9) and after (autumn 2005–autumn 2007; *n* = 21) the first recording of *P. parva* in the Ciemięga River; Min–Max—minimum–maximum, U, Z—the statistics of a Mann–Whitney test.

| Groups and Fish Species | Before Finding *P. parva* | | After Finding *P. parva* | | U | Z | *p* Value |
|---|---|---|---|---|---|---|---|
| | Mean ± SD | Min–Max | Mean ± SD | Min–Max | | | |
| Native fish species | | | | | | | |
| Cyprinidae | 8.4 ± 6.7 | 0.0–35.0 | 23.7 ± 25.3 | 0.0–187.5 | 73.0 | −0.97 | 0.33 |
| Gudgeon (*Gobio gobio*) | 5.9 ± 5.4 | 0.0–35.0 | 3.5 ± 2.5 | 0.0–16.5 | 86.0 | −0.38 | 0.70 |
| Roach (*Rutilus rutilus*) | 1.9 ± 2.2 | 0.0–13.0 | 2.3 ± 2.5 | 0.0–23.0 | 83.0 | −0.52 | 0.60 |
| Tench (*Tinca tinca*) | 0.1 ± 0.1 | 0.0–0.5 | 0.8 ± 0.8 | 0.0–7.0 | 75.0 | −0.88 | 0.38 |
| Bleak (*Alburnus alburnus*) | 0.2 ± 0.3 | 0.0–1.5 | 17.0 ± 26.0 | 0.0–187.5 | 52.0 | −1.92 | 0.05 |
| Spirilin (*Alburnoides bipunctatus*) | 0.2 ± 0.1 | 0.0–0.5 | – | – | – | – | – |
| Sunbleak (*Leucaspius delineatus*) | 0.4 ± 0.5 | 0.0–3.0 | – | – | – | – | – |
| Ide (*Leuciscus idus*) | – | – | 0.1 ± 0.2 | 0.0–1.0 | – | – | – |
| Predatory fish | 4.1 ± 2.8 | 0.0–14.0 | 14.1 ± 11.2 | 0.0–80.5 | 79.0 | −0.70 | 0.48 |
| Brown trout (*Salmo trutta*) | 2.9 ± 2.4 | 0.0–14.0 | 13.7 ± 11.3 | 0.0–80.5 | 68.5 | −1.18 | 0.24 |
| Northen pike (*Esox lucius*) | 0.1 ± 0.2 | 0.0–1.0 | 0.1 ± 0.1 | 0.0–0.5 | 88.0 | 0.29 | 0.77 |
| Eurasian perch (*Perca fluviatilis*) | 1.1 ± 0.4 | 0.0–3.5 | 0.3 ± 0.2 | 0.0–2.0 | 76.0 | 0.84 | 0.40 |
| Other fish | 4.2 ± 3.3 | 0.0–15.5 | 8.3 ± 5.5 | 0.0–48.5 | 68.0 | −1.20 | 0.23 |
| Threespine stickleback (*Gasterosteus aculeatus*) | 2.4 ± 2.5 | 0.0–15.5 | 6.1 ± 5.3 | 0.0–48.5 | 68.5 | −1.18 | 0.24 |
| Stone loach (*Barbatula barbatula*) | 1.8 ± 2.0 | 0.0–6.5 | 2.3 ± 1.5 | 0.0–10.0 | 88.0 | −0.29 | 0.77 |
| Non-native fish species | | | | | | | |
| Topmouth gudgeon (*Pseudorasbora parva*) | – | – | 7.5 ± 9.1 | 0.0–81.0 | – | – | – |
| Gibel carp (*Carassius gibelio*) | 1.7 ± 1.0 | 0.0–8.0 | 1.9 ± 1.9 | 0.0–12.0 | 85.5 | 0.83 | 0.69 |
| Brown bullhead (*Ameiurus nebulosus*) | 0.1 ± 0.1 | 0.0–0.5 | – | – | – | – | – |
| Total (excluding *P. parva*) | 18.6 ± 2.8 | 0.0–35.0 | 48.0 ± 32.5 | 0.0–73.2 | 3.00 | −1.59 | 0.11 |
| Total (including *P. parva*) | 18.6 ± 2.8 | 0.0–35.0 | 55.5 ± 35.5 | 0.0–187.5 | 40.5 | −2.4 | 0.01 * |

\* differences significant at $p \leq 0.05$.

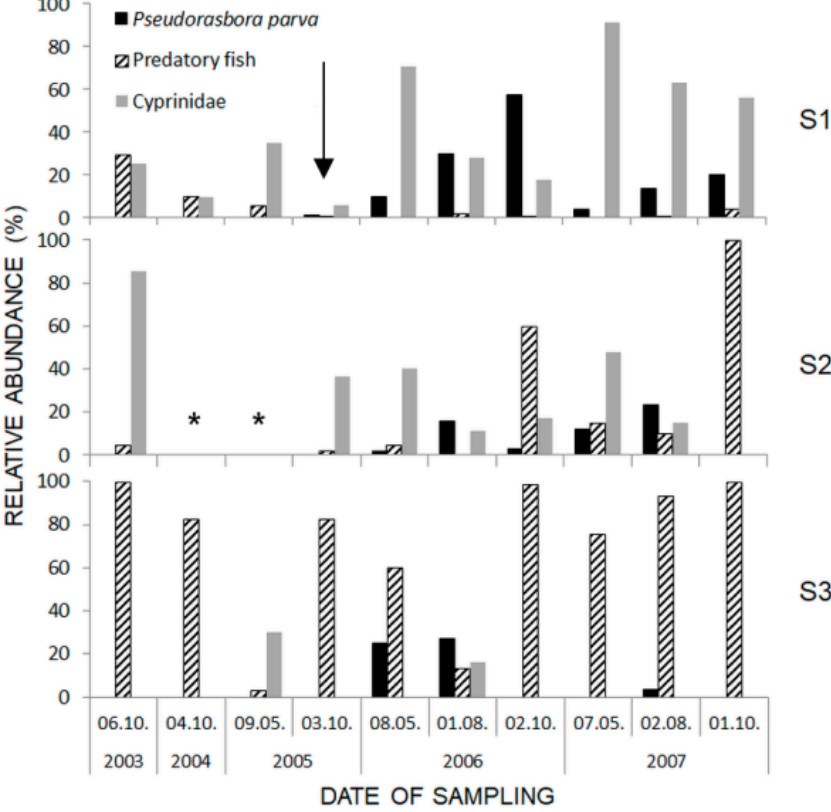

**Figure 2.** The relative abundance (%) of Cyprinidae, predatory fish and *Pseudorasbora parva* in the Ciemięga River (sites S1–S3) in 2003–2007, the arrow indicates the time of appearance of *Pseudorasbora parva* in the Ciemięga river; *—only one species of fish the Stone loach (*Barbatula barbatula*) was noted.

**Table 4.** PERMANOVA results of the impact of site and the presence of *Pseudorasbora parva* on the taxonomic composition of the studied fish community. Df—number of degrees of freedom, SS—sum of squares, MS—mean square, F—test statistic, Pr (>F)—*p*-value (statistical significance).

| | Effect | df | SS | MS | F | Pr (>F) |
|---|---|---|---|---|---|---|
| Two-way analysis | Site | 2 | 2.21 | 1.11 | 4.11 | 0.001 * |
| | *P. parva* | 1 | 0.39 | 0.39 | 1.46 | 0.168 |
| | Site × *P. parva* | 2 | 0.96 | 0.48 | 1.78 | 0.045 * |
| | Residuals | 24 | 6.46 | 0.27 | 0.64 | |
| **Separate 1-way analyses for particular sites** | | | | | | |
| Site 1 | *P. parva* | 1 | 0.46 | 0.46 | 1.72 | 0.093 |
| | Residuals | 8 | 2.14 | 0.27 | 0.82 | |
| Site 2 | *P. parva* | 1 | 0.64 | 0.64 | 2.03 | 0.045 * |
| | Residuals | 8 | 2.51 | 0.31 | 0.80 | |
| Site 3 | *P. parva* | 1 | 0.25 | 0.25 | 1.11 | 0.357 |
| | Residuals | 8 | 1.81 | 0.23 | 0.88 | |

\* differences significant at $p \leq 0.05$.

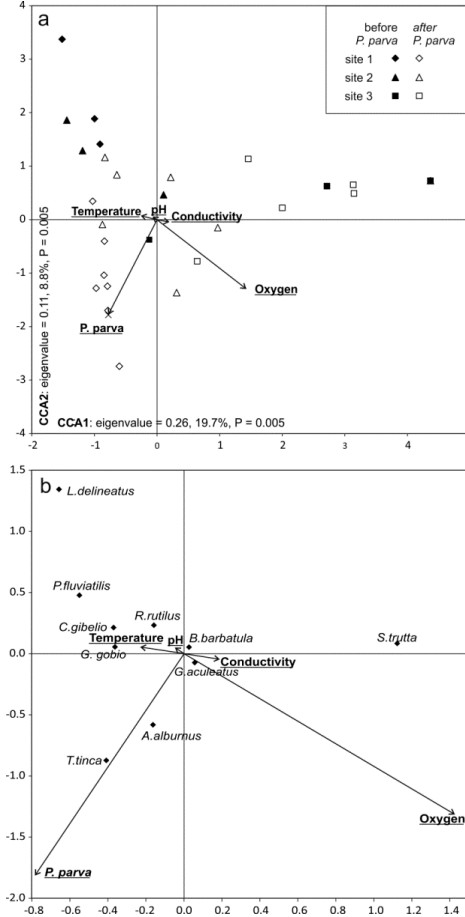

**Figure 3.** Canonical correspondence analysis ordination of the studied samples: (**a**) fish species (**b**) and environmental variables (shown as arrows). The length of an arrow along a particular axis denotes the strength of the correlation between a given environmental variable and that axis. Species and samples are located according to their distribution along environmental gradients denoted by arrows. Species are located near the samples for which they are typical. Filled symbols are pre-invasion and clear symbols are post invasion for S1 (◊), S2 (△), and S3 (□).

**Table 5.** Characteristics of diversity indices (mean, standard deviation (SD), and minimum (Min) and maximum (Max)) before and after *Pseudorasbora parva* was found in the Ciemięga River; U, Z—the statistics of a Mann–Whitney test.

| Indices | Before *Pseudorasbora parva* | | After *Pseudorasbora parva* | | U | Z | *p* Value |
|---|---|---|---|---|---|---|---|
| | Mean ± SD | Min–Max | Mean ± SD | Min–Max | | | |
| Estimated Species richness | 7.5 ± 2.1 | 4.4–9.8 | 10.1 ± 2.0 | 4.4–12.0 | 35.0 | −2.7 | 0.01 * |
| Rarefacted species richness (for 20 individuals) | 3.3 ± 2.1 | 1.0–5.7 | 3.7 ± 1.7 | 1.0–6.2 | 83.0 | −0.5 | 0.60 |
| Margalef index (R) | 1.3 ± 1.4 | 0.0–3.4 | 1.6 ± 1.3 | 0.0–3.6 | 82.0 | −0.6 | 0.57 |

* differences significant at $p \leq 0.05$.

Rarefaction curves (Figure 4) for sites 1 and 3 revealed greater species evenness (steeper curves) for periods before the appearance of *P. parva*, whereas the situation at site 2 was the opposite. The mean fauna turnover rate for all the study sites was 0.20.

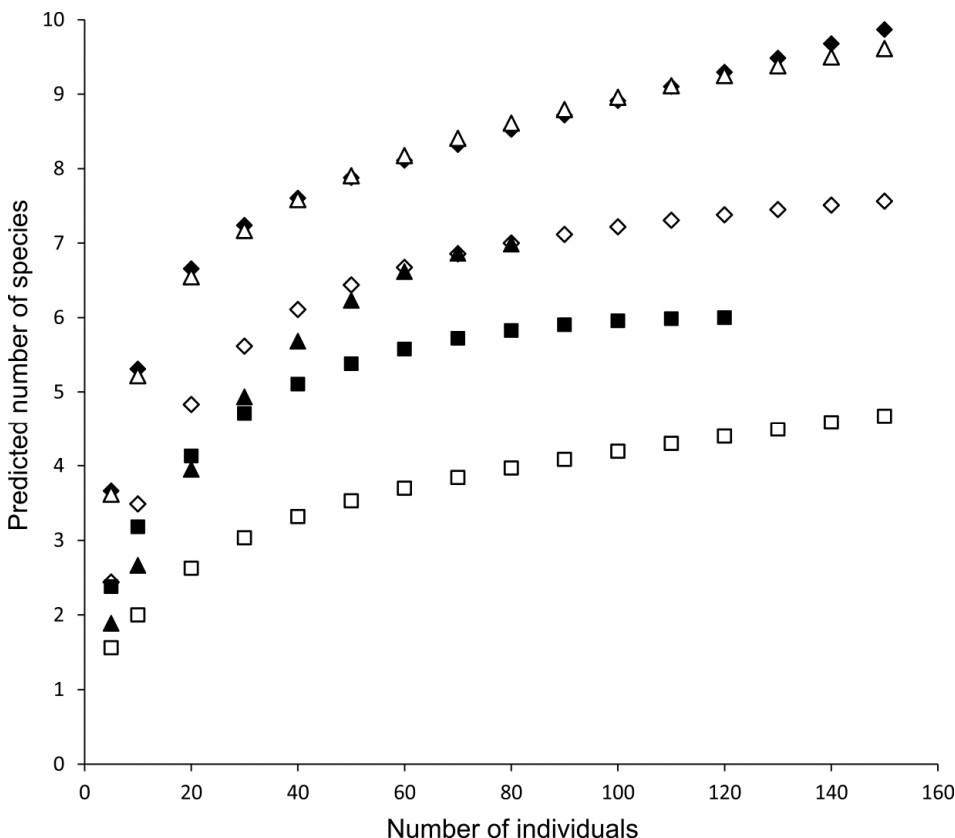

**Figure 4.** Rarefaction curves showing the numbers of species expected for particular numbers of individuals collected. The curve steepness at its initial part indicates the community evenness (the higher steepness, the higher evenness), whereas the curve height shows the number of species. Filled symbols are pre-invasion and clear symbols are post invasion for S1 (◊), S2 (Δ), and S3 (□).

*Pseudorasbora parva* density was strongly and positively correlated with the estimated species richness, rarefacted species richness, and the Margalef diversity index (Table 6). The total number of fish in the river did not vary significantly with the *P. parva* density, but the overall density and proportion of cyprinids were positively correlated with the *P. parva* density (Table 6). A closer analysis of correlations showed that among the species of the Cyprinidae family, the density of *G. gobio*, *T. tinca*, and *A. alburnus* significantly increased with that of *P. parva* (Table 5). Despite a tendency for a positive correlation, the presence of *P. parva* did not significantly affect the *Rutilus rutilus* (L.). A positive and significant correlation was also demonstrated for *Barbatula barbatula*, but not for *G. aculeatus* (Table 6).

The *P. parva* density was negatively correlated with the abundance and density of predatory fish (Table 6). However, among these species, a significant negative correlation with *P. parva* density was noted only for the brown trout, but not for the other predators, such as the perch (Table 6).

**Table 6.** Spearman correlations between *P. parva* density (in catch per unit effort (CPUE)) and selected variables (*n* = 21).

| Variables | r | t (N − 2) | *p* Value |
|---|---|---|---|
| Total density of fish | 0.29 | 1.34 | 0.20 |
| Estimated species richness (eS) | 0.77 | 5.30 | <0.01 * |
| Rarefacted species richness (for 20 ind.) | 0.68 | 4.04 | <0.01 * |
| Margalef index (R) | 0.76 | 5.18 | <0.01 * |
| Density of predatory fish (CPUE) | −0.52 | −2.63 | 0.02 * |
| Abundance of predatory fish (%) | −0.61 | −3.39 | 0.01 * |
| Density of brown trout (CPUE) | −0.62 | −3.10 | 0.01 * |
| Density of Eurasian perch (CPUE) | −0.26 | −1.25 | 0.25 |
| Density of Cyprinidae fish (excluding *P. parva*) (CPUE) | 0.62 | 3.47 | 0.01 * |
| Abundance of Cyprinidae fish (excluding *P. parva*) (%) | 0.49 | 2.46 | 0.02 * |
| Density of gudgeon (CPUE) | 0.67 | 3.94 | <0.01 * |
| Density of roach (CPUE) | 0.28 | 1.28 | 0.21 |
| Density of tench (CPUE) | 0.63 | 3.53 | 0.01 * |
| Density of bleak (CPUE) | 0.64 | 3.59 | 0.01 * |
| Density of threespine stickleback (CPUE) | 0.38 | 1.82 | 0.08 |
| Density of stone loach (CPUE) | 0.48 | 2.42 | 0.03 * |

* correlations significant at $p \leq 0.05$.

## 4. Discussion

Quantification of the impact of *P. parva* on aquatic ecosystems is very difficult and studies about its effect on the fish community after the invasion were rarely dealt. [36]. Rabitsch et al. [17] analysed the occurrence and importance of non-native species in European waters and classified *P. parva* as a potentially invasive species. In addition, *P. parva* is the only species in this group of fish with stable populations in many parts of Europe.

In the present research, the appearance of *P. parva* coincided with the disappearance of several species but did not affect the density of the other fish (Table 3). The differences in fish community composition before and after the appearance of *P. parva* were significant, despite previous research finding no effect on native food webs and ecosystems in rivers [37] and stagnant waters [38]. These ambiguous results indicate the need for further and more insightful observations in order to understand these mechanisms. *P. parva* is important because this invasive species occupies the same habitat as resident species. Observations by Beyer et al. [39] showed that *P. parva* occupied the same habitat as the brown trout *Salmo trutta*, chub *Leuciscus cephalus* (L.), and the stone loach *Barbatula barbatula*, while showing no ecological impact on these species.

Several authors pointed out that that invasive *P. parva* might compete with native fish for food [8,10,24]. However, these data are based only on the suggestion of potential competition rather than direct evidence. These suggestions were partly confirmed only by the observations conducted by Adámek and Sukop [23] in ponds. However, the impact of *P. parva* on native fish species may be more direct. For example, a negative impact on the growth rate of all size classes of *R. rutilus* in small ponds was noted by Britton et al. [38], although there was no clear significant effect on the fish community after the appearance of *P. parva*. Giannetto et al. [40] showed that the assessment of the relative weights of native fish species may be a good indicator of the pressure of non-native fish species on aquatic ecosystems. A higher density of non-native species, including *P. parva*, was negatively correlated with the relative weight of *Telestes muticellus* (Bonaparte 1837) but did not affect the weight of *Barbus tyberinus* (Bonaparte, 1839), *Leuciscus cephalus*, *Leuciscus lucumonis* (Bianco 1983), or *Rutilus rubilio* (Bonaparte 1837).

In the present study, most of the resident species showed positive correlations with the density of *P. parva*. Predatory fish species, especially the brown trout, were the exception. When *P. parva* was present, a several-fold (though statistically insignificant) increase was noted in the density of *A. alburnus*, while the total density of fish increased significantly according to the presence of *P. parva* (Table 3). However, as reported by Gozlan et al. [20] and Britton et al. [41], the impact of *P. parva* on resident fish species may only become evident after about four years.

The absence of sunbleak *Leucaspius delineatus* in ichthyofauna after the appearance of *P. parva* may indicate a clear relationship between these species. Although small numbers of this species were noted before the appearance of *P. parva*, it was not caught at any of the study sites after autumn of 2005. *Pseudorasbora parva* can transmit *Sphaerothecum destruens* [20], inhibiting spawning [42] and potentially leading to the extinction of *Leucaspius delineatus* and a range of other freshwater fish [43]. It is possible that in present study the presence of *P. parva* through its multiple negative impacts that have been previously found on *L. delineatus*, caused its absence after invasion.

The results show that hypothesis (i) (postulating that the appearance of *P. parva* may be associated with changes in the fish community or specific fish species appearances) can be partly supported. The appearance of *P. parva* was not significantly related to the density of particular fish species but could have contributed to the disappearance of two of them. Moreover, it coincided with a significant increase in the total fish number and significant changes in the fish community. This inconclusive relationship may be due to the fact that *P. parva* in the Ciemięga River is in the initial stage of invasion, and it is too early to confirm its impact decisively.

The appearance of a new invasive fish species was not significantly (or even positively) correlated with species richness (Tables 5 and 6). Hypothesis (ii) can be rejected, as the appearance of P. parva was not negatively correlated with fish richness. It seems that ecosystems with a higher species richness and higher diversity indices could more easily defend against the invasion of new species. However, Kennard et al. [37] showed that despite high species richness in the rivers of Australia, these rivers did not show increased resistance to invasion. These authors also suggested that abiotic parameters, often resulting from riverbed modifications and human activity, may be a factor assisting invasions. In the present study, richness indices were relatively low, and the community did not respond to the presence of non-native species. A significant increase was observed only in the case of the estimated species richness (Table 5). While it is true that some invasive species can disrupt ecosystems and eliminate native species, the same processes may also lead to an increase in ecological diversity [44]. Nevertheless, any increase in local biodiversity resulting from the introduction of new species will be offset by a decline in global biodiversity [45]. Therefore, the most rational action is to assess the risk posed by invasive species and to observe whether a given species causes the disappearance of native species or finds its own place in the habitat. It is possible that in the present study the first stage of invasion was observed, and changes have not yet been noticeable.

*Pseudorasbora parva* had little effect on the fish community and diversity indices in the present study, but, as Pinder et al. [4] and Britton et al. [41] have reported, the appearance of this non-native species does not always mean a successful invasion, e.g., due to predation pressure or the absence of an ecological niche.

Hypothesis (iii), concerning the influence of predatory fish on the presence of *P. parva*, was partly confirmed. In the present study, a greater density of predatory fish, mainly *S. trutta*, could have a negative effect on the density of *P. parva*. Other predatory fish species, such as *E. lucius*, were represented by a small number of individuals, while *P. fluviatilis* were too small to exert predation pressure on this non-native fish species (mean *Tl* less than 11 cm). Interactions with *S. trutta* could help control of the numbers of *P. parva*. However, *P. parva* simultaneously depends on the part of the river (the study site). At the site in Pliszczyn (S3), where the highest density of *S. trutta* was noted, *P. parva* was subjected to greater predation pressure. This pressure could result in the reduction of the number of this invasive species. According to Beyer [46], brown trout can feed on *P. parva*, though its share in the predator's diet was only 7%. In addition, this pressure was confirmed by local anglers,

who had observed the presence of *P. parva* in *S. trutta* stomachs (an anonymous angler, pers. comm., July 2007). Moreover, research conducted by Musil and Adámek [26] on the relationship between predatory species and *P. parva* in a small river channel in the Czech Republic showed that this alien species was the basis for the diet of predators. Similarly, the observations by Lemmens et al. [47] provided evidence for the negative effects of pike stocking on the abundance and biomass of *P. parva* in experimental ponds. The results of these authors suggest that the presence of predatory fish in freshwater ecosystems can considerably enhance the biotic resistance of fish communities against the invasion by *P. parva*. Moreover, Beyer et al. [39] found that the presence of small specimens of *P. parva* was strongly associated with large-bodied fish species, such as *S. trutta* and *L. cephalus*. On the other hand, Kapusta et al. [48] reported that predatory species in the ichthyofauna of the heated Lake Licheńskie had no effect on the abundance of this invasive species.

In conclusion, the appearance of *P. parva* can affect the total fish density, fish community composition, and, possibly, in the longer term, result in the displacement of some fish species (e.g., *L. delineatus*). Furthermore, *P. parva* did result in an increase in species richness. Predatory fish species (here, the brown trout) can effectively limit the invasion of this species. Nevertheless, invaded ecosystems should be continuously monitored for changes in their fish communities during subsequent stages of invasion. This will improve o general understanding of the ecology of *P. parva* and provide information on the stability of its population in the newly settled region.

**Funding:** This research received no external funding.

**Acknowledgments:** The author thank P. Mazurek, R. Kostelecki, M. Woźny, and T. Pawluk for helpful assistance in the field work.

**Conflicts of Interest:** The author declares no conflict of interest.

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
