# Peer review of "Changes in a Fish Community in a Small River Related to the Appearance of the Invasive Topmouth Gudgeon Pseudorasbora parva (Temminck & Schlegel, 1846)"

_water, doi:10.3390/w11091857_

Round 1
Reviewer 1 Report
This research has been well carried out and adds to a body of evidence of the impacts of the highly invasive top mouth gudgeon. Several sections could be more concise and less repetitive, have given alternative text to use for many examples. The methods are thorough and this is a well carried out piece of research. The results are mostly well displayed and clearly presented. The discussion is thorough and puts these results into context with existing research. I think the discussion could benefit from extra discussion around the other fish that was not detected post-invasion and what further steps this research could take. See attachment for specific comments for improving the manuscript.

Author Response
Response to Reviewer 1 comments about my manuscript entitled “Did the appearance of the invasive topmouth gudgeon (Pseudorasbora parva (Temminck & Schlegel, 1846)) affect a native fish community in a small river?”
I would like to thank two anonymous Reviewers who objectively assessed my manuscript. I think their comments have improved my paper.
Reviewer 1 (‘water-555593-review.pdf’)
Line 24 - The sentence beginning in line 24 in the abstract has been changed.
Line 29 – comma was added
Line 30 – „here” was removed
Line 31 – "for" to "of" was replaced
Line 33 –keywords „impact on native fish” was changed to „native fish”, „small” was deleted
Line 38 – sentence was changed and deleted unnecessary
Line 40 – sentence was changed
Line 43-46 – paragraph was deleted and sentence was added to previous paragraph
Line 52-54 – paragraph was deleted, sentence was changed and added to previous paragraph
Line 57 – word ‘appearance’ was changed to ‘invasion’
Line 58 „the” and „small river” were removed
Line 60 – word „pest” was replaced to „invasive”
Line 66 – sentence was changed
Line 72 – I decided to leave the reference to Table 1 in a separate sentence, because the previous sentence concerns measurements of physical and chemical parameters, not location and description of study sites
Line 76 Figure 1 was improved
Line 83 – I didn't take into account the comment because the description concerns on the river bottom, not the benthos in the river bottom. The other comments to Table 1 have been included.
Line 147 – phrase „(see the Results)” was deleted
Line 149 – citations were changed and citation regarding package R (ref. no. 35) was added
Line 152 – word ‘individuals’ was remove
Line 160 – excessive citation of 'Table 3' was removed
Line 161 - word order was changed
Line 163-164 – values of CPUE to one decimal place were changed
Line 165 - ‘the’ before gudgeon was removed
Line 165 - phrase was changed
Line 165 – 167 – sentence was verified
Line 171 – 172 all reviewer’s comment included,
Line 172- 173 - The sentence was partly left because my intention was to indicate the fact that the ratio between the Cyprinidae and P. parva changed.
Line 176 - all reviewer's comment included
Line 181 – Figure 3 citation moved to the end of the sentence
Line 182 - 183 - In this analysis, P. parva was an influence factor (see arrow), the result showed negative relationships (they are separated by a vertical axis). The citing of the figure 3 has been made more specific.
188- 193 - excessive citation of Figure 5 was removed
Line 189 - the results were corrected to one decimal place
Table 2 - the title of table 2 has been corrected
Figure 2 was improved and figure captions changed
Table 4 - the title of table 4 has been corrected
Figure 4 - all reviewer's comment included
Table 5 - all reviewer's comment included
Table 6 - comma replaced by a period
Line 266 – word was chcnged
Line 268 – phrase ‘on the grey list of species’ was removed
Line 271- word ‘Moreover’ was removed
Line 272 – 273 – the sentences were merged
Line 279 - the sentence was revised
Line 289 - the names of fish species were cited after source Giannetto et al. [40]
Line 297 - the sentence was revised
Line 299 - paragraph was changed
Line 314 - the sentence was changed
Reviewer 2 Report
In the article «Did the appearance of invasive topmouth gudgeon affect a native fish community in a small river?», the author describes changes in a fish assemblage at three sites of a river in Poland before and after the invasion by topmouth gudgeon.
These kinds of before-after datasets are indeed rare and should be published. Some amendments are however recommended prior to publication. In particular, it is recommendable that the work, to increase its reception, is professionalized. Some suggestions are listed below.
- The title should be formulated as statement. Either there is a change, or there is no change, or there is change at some sites but not others. If it is not possible to make a concise statement about a paper’s content with a “tweet like quality”, it should not be published. There are many online resources to help.
- Structuring the abstract like suggested in the link might help:
https://cbs.umn.edu/sites/cbs.umn.edu/files/public/downloads/Annotated_Nature_abstract.pdf
- Replacing vague statements with specific statements throughout the paper is recommended.
Example:
“Many papers have reported this invader in various countries.”
How many? In which countries?
“In the recent years, a number of review papers have been written on the importance of invasive fish species.”
How many years? How many papers? Importance – in which respect? Which species?
- Replacing “school text” statements with professional statements throughout the paper is recommended.
Example:
“On the other hand, the ecosystem can defend itself against invasive species.” à can be replaced with something along the lines of “Certain ecosystem properties excert protective effects regarding species invasions. For example, …. (ref) in case x, and ….(ref) in case y.”.
- Provide information on the species in the introduction. Origin, role in ecosystem, size, spawning time, common impacts, a picture…
- Voicing a few concice predictions at the end of the introduction is recommendable. “Based on …., we expect to see less invasive individuals at sites with high densities of top predators.” etc. – link up with hypotheses in the discussion.
- Figure 1: More labels. More scales. 1-7 km is a weird unit? Draw the barrier. Also add a timeline of the sampling, for example a time arrow below the figure with marks for samplings. Also add major cities or human impact areas. Mountain area, agricultural area?
- Groups: quite artificial classification in functional and taxonomic units. Find a classification scheme that is more consistent.
- Indices: In each case, state what the selected index displays, how it compares to other indices, and why it was chosen over other indices with the same function (also see Magurran and Henderson 2010 https://doi.org/10.1098/rstb.2010.0285)
- Figure 2: Provide a graph with all species, 15 species is not too many to display. Scale the time axis properly and clearly label similar / different times of year. At present, all timepoints are presented equal, but actually it is a mix of spring and autumn samples and they are not evenly spaced. This should be represented adaequately.
- Figure 3: Explain what can be seen in the legend. Explain the figure so that a person not familiar with CCA (like myself) understands what they see. What do the arrows represent? Same applies to Figure 4.
- Discussion: Some of this may have a better place in the introduction (for example, previous work on P.parva impacts).
- Discussion: Structure with subheadings, for example Role of predators, Role of competitors, …
Good luck with the revisions. And don’t be too much in love with statistical tests and p-values to claim there is or is not a difference… https://www.nature.com/articles/d41586-019-00857-9
Author Response
Response to Reviewer 2 comments about my manuscript entitled “Did the appearance of the invasive topmouth gudgeon (Pseudorasbora parva (Temminck & Schlegel, 1846)) affect a native fish community in a small river?”
I would like to thank two anonymous Reviewers who objectively assessed my manuscript. I think their comments have improved my paper.
Reviewer 2 (file: ‘Revie 2 Water.docx’)
Comment:
- The title should be formulated as statement. Either there is a change, or there is no change, or there is change at some sites but not others. If it is not possible to make a concise statement about a paper’s content with a “tweet like quality”, it should not be published. There are many online resources to help.
Respons: New title was proposed
“Changes in a fish community in a small river related to the appearance of the invasive topmouth gudgeon (Pseudorasbora parva (Temminck & Schlegel, 1846)”
Comment:
- Structuring the abstract like suggested in the link might help:
https://cbs.umn.edu/sites/cbs.umn.edu/files/public/downloads/Annotated_Nature_abstract.pdf
Respons: Thank you for infomation about an interesting document that facilitates writing an abstract.
Comment:
- Replacing vague statements with specific statements throughout the paper is recommended.
Example:
“Many papers have reported this invader in various countries.”
How many? In which countries?
“In the recent years, a number of review papers have been written on the importance of invasive fish species.”
How many years? How many papers? Importance – in which respect? Which species?
Respons: The subject of alien species is quite popular in scientific literature and I think that it is difficult to place and cite all items regarding a problem in every publication. Often, such publications are local in regional journals and are difficult to find them and cite. In the current version, I think that the publications quoted by me are correctly selected and adding more sources will significantly increase the list of references.
Comment
- Replacing “school text” statements with professional statements throughout the paper is recommended.
Example:
“On the other hand, the ecosystem can defend itself against invasive species.” à can be replaced with something along the lines of “Certain ecosystem properties excert protective effects regarding species invasions. For example, …. (ref) in case x, and ….(ref) in case y.”.
Respons: Does this note apply to the whole text? Some of these types of wording have changed (for example line 55- 58)
Comment:
Groups: quite artificial classification in functional and taxonomic units. Find a classification scheme that is more consistent.
Respons: I don't really understand whether it concerns tables, figures or other analyzes. When describing the tables and analyzes, I tried to best classify fish species so that the results could be interpreted and described as best as possible.
Comment:
- Provide information on the species in the introduction. Origin, role in ecosystem, size, spawning time, common impacts, a picture…
Respons: There are several publications on P. parva and most of this information is included. I believe that the lack of this information in my work is not a mistake. In addition, this data was included in my another paper on this species(see: https://www.aiep.pl/volumes/2010/2_3/pdf/07_1110_P3.pdf)
Comment:
Indices: In each case, state what the selected index displays, how it compares to other indices, and why it was chosen over other indices with the same function (also see Magurran and Henderson 2010 https://doi.org/10.1098/rstb.2010.0285)
Respons: In my opinion, I chose the best indicators for this type of analysis. These indicators are the result of several parameters and sometimes indicate difficult to interpret values. Sometimes the only option is to compare the values of individual indexes with each other. Thank you for the link to an interesting source that I will certainly refer to for this type of research.
Figure 2 comment:
Respons: Figure 2 has been changed mainly according to the comments of Reviewer 1 and partly Reviewer 2. I believe that adding all 15 species lines in the figure would make it less readable.
Figure 3 and 4 comments
Respons: Figures was improved and descriptions was added to figure legends
Comment to discussion:
Respons: In my opinion the adding subheadings will not be a good idea. Some of the issues raised in the discussion are too modest or cannot be attributed to any subheading. I left the discussion structure unchanged.
Round 2
Reviewer 2 Report
Since the authors provided scarce comments and did not include original comments nor details on how they were adressed in their response, judging the quality of the revisions is challenging.
Some aspects have improved, such as Figure 1, or more specific and more scientifically exact phrasing, or an attempt to explain CCA in the Figure 3 legend. However, several relevant comments have not been adressed, for example questions regarding the rationale of species grouping ("Predators, Cyprinids, other fish"), or regarding the rationale of species richness indices choice, or regarding the display of non-linearly distributed timepoints in a linear fashion in Figure 2.
I don't intend to be mean, and I am aware that this is likely an important publication. However, it is a responsibility of peer review to ensure the scientific quality and the correct interpretation of data of manuscripts. The authors have not been able to do a convincing job on either of the two aspects.
I will not consider the manuscript acceptable at the present timepoint without revisions. Also, considering the author's reluctancy to implement relevant suggestions and their disregard for the reviewers' time, as exemplified by the lack of detail in the response, I am not available as reviewer for another round of revisions, should it take place.
Author Response
Respons to Reviewer 2 comments (round 2)
Manuscript ID - water-555593
Title: Did the appearance of the invasive topmouth gudgeon (Pseudorasbora parva (Temminck & Schlegel, 1846)) affect a native fish community in a small river?
New title:
“Changes in a fish community in a small river related to the appearance of the invasive topmouth gudgeon (Pseudorasbora parva (Temminck & Schlegel, 1846)”
Respons to Reviewer 2 comment:
Comment:
Since the authors provided scarce comments and did not include original comments nor details on how they were adressed in their response, judging the quality of the revisions is challenging.
Respon to comment: With the previous corrected version was accompanied text with response to most of Reviewer 2 comments.
Below I present the content of these answers from the previous (first) review.
After first review
Response to Reviewer 2 comments about my manuscript entitled “Did the appearance of the invasive topmouth gudgeon (Pseudorasbora parva (Temminck & Schlegel, 1846)) affect a native fish community in a small river?”
I would like to thank two anonymous Reviewers who objectively assessed my manuscript. I think their comments have improved my paper.
Reviewer 2 (file: ‘Revie 2 Water.docx’)
Comment:
- The title should be formulated as statement. Either there is a change, or there is no change, or there is change at some sites but not others. If it is not possible to make a concise statement about a paper’s content with a “tweet like quality”, it should not be published. There are many online resources to help.
Respons: New title was proposed
“Changes in a fish community in a small river related to the appearance of the invasive topmouth gudgeon (Pseudorasbora parva (Temminck & Schlegel, 1846)”
Comment:
- Structuring the abstract like suggested in the link might help:
https://cbs.umn.edu/sites/cbs.umn.edu/files/public/downloads/Annotated_Nature_abstract.pdf
Respons: Thank you for infomation about an interesting document that facilitates writing an abstract.
Comment:
- Replacing vague statements with specific statements throughout the paper is recommended.
Example:
“Many papers have reported this invader in various countries.”
How many? In which countries?
“In the recent years, a number of review papers have been written on the importance of invasive fish species.”
How many years? How many papers? Importance – in which respect? Which species?
Respons: The subject of alien species is quite popular in scientific literature and I think that it is difficult to place and cite all items regarding a problem in every publication. Often, such publications are local in regional journals and are difficult to find them and cite. In the current version, I think that the publications quoted by me are correctly selected and adding more sources will significantly increase the list of references.
Comment
- Replacing “school text” statements with professional statements throughout the paper is recommended.
Example:
“On the other hand, the ecosystem can defend itself against invasive species.” à can be replaced with something along the lines of “Certain ecosystem properties excert protective effects regarding species invasions. For example, …. (ref) in case x, and ….(ref) in case y.”.
Respons: Does this note apply to the whole text? Some of these types of wording have changed (for example line 55- 58)
Comment:
Groups: quite artificial classification in functional and taxonomic units. Find a classification scheme that is more consistent.
Respons: I don't really understand whether it concerns tables, figures or other analyzes. When describing the tables and analyzes, I tried to best classify fish species so that the results could be interpreted and described as best as possible.
Comment:
- Provide information on the species in the introduction. Origin, role in ecosystem, size, spawning time, common impacts, a picture…
Respons: There are several publications on P. parva and most of this information is included. I believe that the lack of this information in my work is not a mistake. In addition, this data was included in my another paper on this species (see: https://www.aiep.pl/volumes/2010/2_3/pdf/07_1110_P3.pdf)
Comment:
Indices: In each case, state what the selected index displays, how it compares to other indices, and why it was chosen over other indices with the same function (also see Magurran and Henderson 2010 https://doi.org/10.1098/rstb.2010.0285)
Respons: In my opinion, I chose the best indicators for this type of analysis. These indicators are the result of several parameters and sometimes indicate difficult to interpret values. Sometimes the only option is to compare the values of individual indexes with each other. Thank you for the link to an interesting source that I will certainly refer to for this type of research.
Figure 2 comment:
Respons: Figure 2 has been changed mainly according to the comments of Reviewer 1 and partly Reviewer 2. I believe that adding all 15 species lines in the figure would make it less readable.
Figure 3 and 4 comments
Respons: Figures was improved and descriptions was added to figure legends
Comment to discussion:
Respons: In my opinion the adding subheadings will not be a good idea. Some of the issues raised in the discussion are too modest or cannot be attributed to any subheading. I left the discussion structure unchanged.
Reviewer 2 Comments and Suggestions (round 2)
Since the authors provided scarce comments and did not include original comments nor details on how they were adressed in their response, judging the quality of the revisions is challenging.
Some aspects have improved, such as Figure 1, or more specific and more scientifically exact phrasing, or an attempt to explain CCA in the Figure 3 legend. However, several relevant comments have not been adressed, for example questions regarding the rationale of species grouping ("Predators, Cyprinids, other fish"), or regarding the rationale of species richness indices choice, or regarding the display of non-linearly distributed timepoints in a linear fashion in Figure 2.
I don't intend to be mean, and I am aware that this is likely an important publication. However, it is a responsibility of peer review to ensure the scientific quality and the correct interpretation of data of manuscripts. The authors have not been able to do a convincing job on either of the two aspects.
I will not consider the manuscript acceptable at the present timepoint without revisions. Also, considering the author's reluctancy to implement relevant suggestions and their disregard for the reviewers' time, as exemplified by the lack of detail in the response, I am not available as reviewer for another round of revisions, should it take place.
Respons
In the given to me short period of time that I got to response to the Reviewers' comments, I tried to answer most of them and adapt the new version to the comments of both Reviewers. It wasn’t my intention to ignore the work of any Reviewers and I am very sorry that not all my answers were comprehensive and did not satisfy of the Reviewer.
Below are responses to Reviewer 2 comments in the paragraph:
„Some aspects have improved, such as Figure 1, or more specific and more scientifically exact phrasing, or an attempt to explain CCA in the Figure 3 legend. However, several relevant comments have not been adressed, for example questions regarding the rationale of species grouping ("Predators, Cyprinids, other fish"), or regarding the rationale of species richness indices choice, or regarding the display of non-linearly distributed timepoints in a linear fashion in Figure 2.”
Response to:
… questions regarding the rationale of species grouping ("Predators, Cyprinids, other fish"), …
The main assumption for the division of fish into this groups was the type of potential interaction between these groups of fish and P. parva. Predatory fish (perch, pike, brown trout) can eat the P. parva. The different relationship is between this alien species and other cyprinids fish found in this river (ie. competition for food and habitat). Earlier I thought that the description given in the "Materials and Methods" chapter is sufficient (in lines 122-128 of the current version of the document).
Response to:
…. regarding the rationale of species richness indices choice
In my publication I used several indicators that seem to be one of the basic in the study of assessing the biodiversity of fish assemblies. Species richness (estimated as the number of species) was used to analyse the number of species at the studied sites. Additionally, the rarefaction technique was applied to the number of species to take into account differences in abundances, which can affect richness estimates. The Margalef index is another measure of species richness, which was included in the study to make it comparable to similar papers, as it is commonly used in the scientific literature (and also in assessment of biodiversity for the purpose of environmental monitoring and protection) (for example: Hossain et al., 2012, Ding et al. 2018, Wu et al. 2018). In addition, in analysis I also used the turnover rate of fish index (t). This is an indicator that can potentially indicate to what extent the fish groups have changed under the influence of the factor, which in my publication was an alien fish species - P. parva.
I have to admit, that given by Reviewer both the literature source (Magurran and Henderson 2010 https://doi.org/10.1098/rstb.2010.0285) and the indicators described in it (mean rank shift (MRS) and Bray – Curtis dissimilarity) were unknown to me. In my publication, I tried to use indices known to me and previously used in many other studies. Moreover, due to the very short time to responds to the comments of two Reviewers, the application of these new methods of analysis (which I didn't know) proved difficult.
Hossain M.S., Das N.G., Sarker S., Rahman M.Z., 2012. Fish diversity and habitatrelationship with environmental variables at Meghna River estuary, Bangladesh. Egypt. J. Aquat. Res. 38, 213–226. https://doi.org/10.1016/j.ejar.2012.12.006
Ding Y., Wu Z.Q., Zhu Z. J., Yan J. 2018. Species composition, trend biodiversity variation and conservation of the fish in Lijiang River (in China). Environ. Biol. Fishes, 101, 675–685. https://doi.org/10.3390/su11041135
Wu Z.-Q., Zou Q., Chang T., Zhang D., Huang L.-L., 2018. Seasonal dynamics of the juvenile fish community structure in the Maowei Sea mangroves. PLoS ONE 13(2): e0192426. https://doi.org/10.1371/journal.pone.0192426
Response to:
… regarding the display of non-linearly distributed timepoints in a linear fashion in Figure 2.
This figure 2 was well rated by "Reviewer 1", but I must admit that the comment concerning about "... the display of non-linearly distributed timepoints in a linear fashion ..." is well founded. The figure 2 was improved.